# Development of a core outcome set for evaluative research into paediatric cerebral visual impairment (CVI), in the UK and Eire

Anna Pease ![ORCID],[1] Trudy Goodenough,[1] Cath Borwick,[1] Rose Watanabe,[1] Christopher Morris ![ORCID],[2] Cathy Williams ![ORCID] [1]

[1]Bristol Medical School, University of Bristol, Bristol, UK
[2]University of Exeter Medical School, Exeter, UK

**Correspondence to**
Dr Cathy Williams;
cathy.williams@bristol.ac.uk

## ABSTRACT

**Objectives** Cerebral visual impairment (CVI) comprises a heterogeneous group of brain-related vision problems. A core outcome set (COS) represents the most important condition-specific outcomes according to patients, carers, professionals and researchers. We aimed to produce a COS for studies evaluating interventions for children with CVI, to increase the relevance of research for families and professionals and thereby to improve outcomes for affected children.

**Design** We used methods recommended by the Core Outcome Measures in Effectiveness Trials Initiative. These included a proportionate literature review of outcomes used in previous studies; qualitative interviews with children and families; a two-round Delphi survey involving parents, children and professionals and a consensus meeting to ratify the most important outcomes.

**Setting** Telephone interviews and online Delphi surveys of participants who all lived in UK or Eire.

**Participants** Eighteen parents and six young people were interviewed. Delphi participants (n=80 did both rounds) included professionals working with children who have CVI (teachers, orthoptists, ophthalmologists, optometrists, qualified teachers for visually impaired, family members (parents and siblings) and affected children.

**Results** The literature review included 13 studies yielding 37 outcomes. Qualitative interviews provided 22 outcomes. After combining and refining similar items, the first round contained 23 outcomes and the second 46. At the consensus meeting, 5 attendees recommended 27 outcomes for inclusion in the CVI COS, of which 15 were ratified as most important, including 4 related to vision; 1 to family well-being; 1 to adults around the child being informed about CVI and the rest to the child's abilities to engage with people and surroundings.

**Conclusions** Good engagement from participants led to the development of a COS. Future research will be useful to identify the best ways to measure COS items and potentially to update this COS as more interventions for CVI are developed.

**Trial registration number** ISRCTN13762177.

## STRENGTHS AND LIMITATIONS OF THIS STUDY

⇒ Robust tried and tested methods advocated by the Core Outcome Measures in Effectiveness Trials Initiative were used to propose a core outcome set (COS) for evaluative trials and routine practice in hospitals and schools on childhood cerebral visual impairment (CVI).
⇒ Parents, children and professionals in the UK and Eire took part in the process as interviewees, participants in a Delphi survey and/or consensus meeting and suggested several of the outcomes included in the final COS.
⇒ The inclusion of a self-selecting sample means the precise criteria used to diagnose the participating children with CVI were not sought which may limit the generalisability of the findings, as well as the location being restricted to the UK and Eire.
⇒ The COS is based on ratings by children, families and professionals, therefore, the core outcomes identified may not reflect the views of researchers or journal editors who were not involved.
⇒ Several outcomes were broad concepts and further research is needed to determine how to measure the COS.

of vision.[1–3] Although there is not yet an agreed definition for CVI, a recent review has suggested that CVI be regarded as a verifiable impairment of vision that is not attributable to ocular or anterior visual pathway (optic nerve) disorders.[4] However, different groups and centres take different views on which visual disorders can be described as CVI[3 5] and, in particular, there is variation in the levels of visual acuity loss that different authors describe as consistent with CVI.[4] There are not at present and nationally or internationally agreed guidelines for the management of CVI in children and clinical practices vary according to location, both within and between countries.

A core outcome set (COS) represents the most important condition-specific outcomes

## INTRODUCTION

Cerebral visual impairment (CVI) refers to a range of brain-related impairments

that reflect the priorities of patients, carers, professionals and researchers. Further work is then required to establish 'how' to measure the items in the COS. Use of a COS ensures research findings are relevant to health service users, that is, patients and carers, as well as to healthcare professionals. A COS facilitates synthesis across studies and aggregation of data in meta-analyses or evidence syntheses, to develop a robust evidence base.[6]

Guidelines exist to aid researchers in developing COS::Core outcome Set-STAndards for development (CO-STAD[7]) and reporting processes: Core outcome Set-STAndards for reporting (CO-STAR[8]), which we have followed insofar as our resources allowed. The Core Outcome Measures in Effectiveness Trials (COMET) database describes established and ongoing COS and did not previously include a COS for CVI. The COMET recommended methods[9] include a literature review to discover which outcomes have already been measured in relevant studies, qualitative research to elicit the views of patients, families and carers; a Delphi survey (an online, iterative survey with anonymous participants[10]) in which stakeholders from patient and professionals groups rated outcomes by importance and finally a consensus meeting involving representatives of all stakeholder groups to discuss and ratify the final items in the COS. These methods have recently been used to produce COS the scope of which was the assessment of brain-related impairments of vision in adults who have suffered a stroke.[11] We aimed to produce a COS with a scope of studies evaluating interventions for children with CVI, so as to enhance the relevance for patients of future intervention studies and ultimately improve outcomes for children with CVI, rather than aiding professionals with diagnosis.

## METHODS
### Scope
This COS relates to children aged up to 18 years in the UK and Eire, and to interventions to help children with CVI in any setting, including evaluative trials and routine practice in hospitals and schools. We took an inclusive approach and included CVI as a broad concept, diagnosed by a relevant professional, to reflect (1) the current lack of consensus about which brain-related impairments should be included and (2) variation in availability of detailed vision testing to elicit all the potential manifestations of CVI.

The study was registered with the COMET Initiative (http://www.comet-initiative.org/studies/details/1032). Written consent was documented for the interviews, from both parents and young people. Consent to the Delphi survey was implied by the completion and return of the questionnaires.

### Patient and public involvement
We used a range of methods to involve patients and the public and these are summarised in figure 1, using the GRIPP2 recommended headings.[12] We convened

meetings with two group of families: one in the Bristol area whose children have CVI and the other with the PenCRU Family Faculty at University of Exeter Medical School. The Peninsula Childhood Disability Research Unit (PenCRU) Family Faculty comprises parents whose children have various neurodevelopmental conditions and who regularly contribute to and advise on research studies. We sought their advice on practical details of interacting with children with neurodevelopmental conditions in the planned qualitative interviews. We presented our plans at the yearly meeting of a family support group for children with CVI (the CVI Society, www.cvisociety.org.uk) and families were invited to participate in the qualitative interviews or Delphi survey and/or invite other people to participate. Study details were posted on the CVI Society website and in their closed Facebook group.

We met with our Professionals Advisory Group which includes education and health professionals with an interest in CVI. They advised on the best ways to contact professionals, for example, by using dedicated email lists. Parents and three young people with CVI then advised on the wording of the outcomes included in the Delphi survey.

### Stage 1: literature review
A proportionate review of the literature was carried out to identify outcomes that have been measured in evaluations of interventions to help children with diagnosed CVI or with brain-related vision impairments if the term CVI was not used. A search strategy was updated from a previous scoping review on interventions for CVI,[13] and the resulting set of papers was searched specifically for papers reporting controlled studies, before-and-after studies or service evaluations, involving children with CVI using any definition or brain-related vision problems. Searches were completed in Medline, Embase, PsycINFO and the Cochrane database of systematic reviews (search dates 1946–May 2018). Papers that reported on eligible studies involving children up to 18 years were included, including papers reporting methods before results were obtained, since these contained outcomes relevant to the purpose of our review. Papers where it was not possible to separate adult from child data, where there was no description of the visual behaviour used to define the child as having CVI or brain-related vision problems and papers not in English (as no translation funds were available), were excluded.

A data extraction form was used to record all outcomes used in each included paper, whether primary or secondary, the instruments used, the study type and setting, participant characteristics and data completion for each outcome. Also included was which domain the outcome mapped to, using the COMET taxonomy.[14] Data were independently extracted by two reviewers (CW and TG) and consensus was reached by discussion with a third reviewer.

**How was the development of the research question and outcome measures informed by patients' priorities, experience, and preferences?**
Parents and patients told us about the wide-ranging spectrum of effects that they felt CVI had on their children's lives. The emphasised the importance of poor mental health for children with CVI but no support.

**How did you involve patients in the design of this study?**
Parents approved the approach recommended by the Comet Initiative and recommended including home-schooled children if possible.

**Were patients involved in the recruitment to and conduct of the study?**
Parents recruited interviewees and Delphi participants via word-of-mouth using a closed Facebook group and parent networks. Parents recommended ways to engage with child interviewees. Parents and children reviewed and revised the wording of the Delphi surveys.

**How will the results be disseminated to study participants?**
We will provide a summary of the process and the findings from the qualitative interviews, Delphi process and consensus meeting to all participants. We will present the findings to the yearly meeting of a parent support group (www.cvisociety), post a summary on our study website, present the findings at conferences and submit a paper to a peer reviewed journal.

**Figure 1** Box describing how parents and children were involved in the study. CVI, cerebral visual impairment.

### Stage 2: qualitative interviews with families where a child has CVI

Participants were children and young people (CYP) aged 6–18 years with a diagnosis (by a professional) of CVI, using any definition and/or parents or carers of children with this diagnosis. To include a range of ages and physical capabilities, maximum variation purposive sampling was used to recruit families to a matrix including younger (6–11 years old) and older (12–18 years old) children, and those with and without a diagnosis of cerebral palsy. Families were recruited from three sources: from local Specialist teachers for Vision Impairment; from the West of England School and College, Exeter (WESC Foundation) Specialist Centre for Visual Impairment, and from the CVI Society, a national parent support group for families of children with CVI (www.cvisociety.org.uk) as described above. Recruitment was also facilitated by snowballing information to other interested groups via from the initial contacts, including links to 'Moorvision' (https://www.moorvision.org/) via a local specialist school (the WESC Foundation specialist school for visual impairment).

Interviews were carried out in person either face-to-face or by telephone or video link. Parents were present for interviews with young people. All interviews were audio-recorded with consent. Topic guides covered what day-to-day family life living with CVI is like, including school, home, family and health and well-being; practicalities of living with CVI; support received and what would have helped in the past and what would help in the future. For interviews with young people, a range of creative activities were used to help elicit their views about what matters most to them, including drawing, colouring and tablet-based art activities.

The interviews were transcribed verbatim and systematic coding across the transcripts was used to identify outcomes for inclusion in the Delphi Survey. The full thematic analysis of these interviews exploring the impact of CVI on the daily lives of children and their families will be reported separately.

### Stage 3: Delphi process

A Delphi survey[10] was used for participants to rank the importance of the outcomes assembled from the systematic review and the interview data. There were two rounds to the survey and a subsequent face-to-face consensus meeting to ratify agreement on the most important outcomes.

Two groups of participants were invited to take part in the Delphi survey: professionals and families. Professionals included ophthalmologists, optometrists, orthoptists, paediatricians and specialist teachers (qualified teachers for visual impairment, QTVIs). Families included parents of children with CVI and children living with the

**Table 1** Outcomes rated in rounds 1 and 2 of the Delphi survey of outcomes to include in a core outcome set for childhood CVI

| Rounds | Outcome | Description given in survey | COMET taxonomy domain |
|---|---|---|---|
| 1&2 | Ability to see things at a distance of at least 3 metres | For example, looking at an eye chart with letters or pictures, with glasses if appropriate, visual acuity | Eye outcomes |
| 1&2 | Ability to focus on near things | For example, toys, people's faces, letters or pictures, with glasses if appropriate | Eye outcomes |
| 1&2 | Ability to keep both eyes together on a near target | For example, when reading, doing artwork, or using a computer or tablet | Eye outcomes |
| 1&2 | Ability to recognise visual targets | For example, in picture books or on worksheets, adapted as necessary | Eye outcomes |
| 1&2 | Ability to track moving objects | For example, follow the movements of other children, or animals or cars or balls | Eye outcomes |
| 1&2 | Mental health | For example, anxiety, depression, behavioural problems, self harm | Mental health including psychiatric problems |
| 1&2 | Child's Emotional well-being | For example, child's level of confidence, happiness etc | Emotional functioning/well-being |
| 1&2 | Child's self-perception | For example, how child thinks of themselves, self-esteem | Emotional functioning/well-being |
| 1&2 | Sleep | Sleeping during the night or at appropriate times | General physiological/clinical |
| 1&2 | Tiredness/fatigue | For example, during the day, not just at night; lack of stamina, whatever the cause | General physiological/clinical |
| 1&2 | Mobility | For example, child able to move around their physical environment, including physical obstacles, within the limits of any motor impairment or disability | Physical functioning |
| 1&2 | Self-care | For example, washing, feeding and dressing, both practical and social aspects | Physical functioning |
| 1&2 | Independence | Able to go out and about and/or carry out activities of daily living appropriate to age and development, in and out of the home | Physical functioning |
| 1&2 | Safety | Awareness of danger (physical and social), appropriate to age or development, in and out of the home | Physical functioning |
| 1&2 | Toileting | As appropriate to physical abilities, including physical and social aspects | Physical functioning |
| 1&2 | Route-finding | Finding way around in the community and in both familiar and new settings, for example, school | Physical functioning |
| 1&2 | Communication | For example, with family, friends and others; verbal and non-verbal | Social functioning |
| 1&2 | Community and social life | For example, taking part in clubs, sport, parties, other leisure activities or entertainments | Social functioning |
| 1&2 | Behaviour | For example, managing emotions at home, school and in the wider community, self regulation | Social functioning |
| 1&2 | Play | Ability to play, be part of playful activities, on own or with others | Social functioning |
| 1&2 | Learning | Whether at school or home-taught, learning and applying knowledge, attention and concentration | Cognitive functioning |
| 1&2 | Financial costs | For example, costs incurred by parents; working time lost | Societal/carer burden |
| 1&2 | Family well-being | For example, parental emotional well-being, family functioning | Societal/carer burden |
| 2 only | Consistency of visual performance | For example, changes in ability to process visual information when tired, throughout the day | Eye outcome |
| 2 only | Functional 3D skills | For example, making models in Lego, or craft modelling, doing jigsaw puzzles, dressing and feeding themselves, if appropriate | Eye outcomes |
| 2 only | Ability to see colour | | Eye outcomes |
| 2 only | Sensitivity to light | For example, may need additional lighting or reduced lighting, or tinted glasses | Eye outcomes |

Continued

**Table 1** Continued

| Rounds | Outcome | Description given in survey | COMET taxonomy domain |
|---|---|---|---|
| 2 only | Ability to deal with crowded or cluttered scene | For example, finding clothes, toys and people in cluttered or crowded spaces | Eye outcomes |
| 2 only | Ability to watch TV, films or run computer games | | Eye outcomes |
| 2 only | Hearing | Being aware of sounds | General physiological/clinical |
| 2 only | Auditory understanding | For example, knowing the direction of sound | General physiological/clinical |
| 2 only | Appreciation/enjoyment of music | Child enjoys listening to/experiencing music in whatever form | Social functioning |
| 2 only | Pain | For example, discomfort due to positioning, headaches or other sources of pain or discomfort | General physiological/clinical |
| 2 only | Child's understanding of own needs and their condition | For example, knowing how CVI and other comorbidities affect them and what helps or hinders their daily lives, including being aware of visual field loss | Emotional functioning/well-being |
| 2 only | Child's resilience | For example, child uses appropriate coping mechanisms or strategies to manage CVI and other conditions | Emotional functioning/well-being |
| 2 only | Reading or accessing books | | Cognitive functioning |
| 2 only | Ability to cope with moving environment | For example, car travel, train travel, bus travel | Physical functioning |
| 2 only | Environmental adaptations | Appropriate adaptations available including: learning materials, support with travel and transport | Physical functioning |
| 2 only | Environmental adaptations are acceptable to the child | Child can or will access/use environmental adaptations as provided, for example, learning materials, support with travel and transport | Physical functioning |
| 2 only | Managing in crowded environments | For example, in shops, cinemas, streets | Social functioning |
| 2 only | Social life/relationships | For example, ability to make and sustain friendships and relationships | Social functioning |
| 2 only | General health | Self or proxy rated health, for example, 'how are you in yourself?' | General physiological/clinical |
| 2 only | Relevant adults aware of CVI | Teachers, family etc know what CVI is and what helps the child | Social functioning |

COMET, Core Outcome Measures in Effectiveness Trials; CVI, cerebral visual impairment; 3D, 3 dimensional; TV, television.

condition. Participants were recruited via email lists for local and national organisations, local schools, specialist teachers and patient support groups who advertised the survey to professionals and families. We also invited those families who took part in the qualitative work to take part in the Delphi survey if they wished, but this was not expected or mandatory. Professionals who responded to flyers or email invitations were sent formal invitations to the survey with a link for completion online.

DelphiManager (a web-based Delphi survey data management system developed by COMET) was used to facilitate the construction, administration and analysis of the surveys. The first round of the survey presented the initial list of outcomes compiled using the literature review and interviews. Respondents were asked to rank the importance of measuring each outcome in CVI research using a 9-point scale, where 9 was 'critical to include' and 1 was 'not important at all'. We also invited respondents in the first round to add any additional outcomes that they felt should be included.

In the second round, respondents were shown the median ranking and range of rankings by each group, for each outcome and were reminded of their own ranking. They were given the opportunity to rank again items from the first round, and the additional items that had been suggested.

### Stage 4: consensus workshop

A workshop was held with attendees from both groups in Bristol, to ratify agreement on the most highly-ranked outcomes. We presented the three lists of outcomes: consensus 'in' with more than 70% from both groups ranking them as 7–9, consensus 'out' with more than 70% from both groups ranking them as 1–3, and partial or no agreement with less than 70% from either group ranking them as 7–9. Attendees discussed the inclusion or exclusion of the partial or no agreement list first, then moved

**Table 2** Results of Delphi survey: percentages of 85 respondents who ranked the outcome with a score of 7–9 indicating they were important to include in the core outcome set for childhood CVI

| Outcome | Domain | Families N=30 | Professionals N=55 |
|---|---|---|---|
| Outcomes in first round | | | |
| Ability to see things at a distance of at least 3 m | Eye outcomes | 68% | 67% |
| Ability to focus on near things | Eye outcomes | 87% | 87% |
| Ability to keep both eyes together on a near target | Eye outcomes | 62% | 53% |
| Ability to recognise visual targets | Eye outcomes | 97% | 95% |
| Ability to track moving objects | Eye outcomes | 97% | 76% |
| Mental health | Mental health | 77% | 83% |
| Child's emotional well-being | Emotional functioning/well-being | 83% | 100% |
| Child's self-perception | Emotional functioning/well-being | 77% | 87% |
| Sleep | General physiological/clinical | 73% | 76% |
| Tiredness/fatigue | General physiological/clinical | 87% | 85% |
| Mobility | Physical functioning | 93% | 81% |
| Self-care | Physical functioning | 80% | 69% |
| Independence | Physical functioning | 83% | 89% |
| Safety | Physical functioning | 97% | 91% |
| Toileting | Physical functioning | 60% | 54% |
| Route-finding | Physical functioning | 77% | 65% |
| Communication | Social functioning | 94% | 96% |
| Community and social life | Social functioning | 83% | 87% |
| Behaviour | Social functioning | 77% | 72% |
| Play | Social functioning | 80% | 87% |
| Learning | Cognitive functioning | 94% | 88% |
| Financial costs | Societal/carer burden | 53% | 41% |
| Family well-being | Societal/carer burden | 76% | 90% |
| Outcomes suggested by participants in round 1 | | | |
| Consistency of visual performance | Eye outcomes | 80% | 63% |
| Functional 3D skills | Eye outcomes | 78% | 53% |
| Ability to see colour | Eye outcomes | 33% | 26% |
| Sensitivity to light | Eye outcomes | 73% | 63% |
| Ability to deal with crowded or cluttered scene | Eye outcomes | 83% | 77% |
| Ability to watch TV, films or run computer games | Eye outcomes | 57% | 39% |
| Hearing | General physiological/clinical | 80% | 68% |
| Auditory understanding | General physiological/clinical | 94% | 68% |
| Appreciation/enjoyment of music | General physiological/clinical | 50% | 41% |
| Pain | General physiological/clinical | 76% | 72% |
| Child's understanding of own needs and their condition | Emotional functioning/well-being | 64% | 68% |
| Child's resilience | Emotional functioning/well-being | 83% | 81% |
| Reading or accessing books | Eye outcomes | 63% | 54% |
| Ability to cope with moving environment | Eye outcomes | 94% | 72% |
| Environmental adaptations | Physical functioning | 87% | 73% |
| Environmental adaptations are acceptable to the child | Physical functioning | 89% | 87% |
| Managing in crowded environments | Social functioning | 80% | 63% |
| Social life/relationships | Social functioning | 87% | 90% |

**Table 2** Continued

| Outcome | Domain | Families N=30 | Professionals N=55 |
|---|---|---|---|
| General Health | General physiological/clinical | 73% | 67% |
| Relevant adults aware of CVI | Delivery of care | 94% | 90% |

Green= 70% or more of each group rated outcome as 7–9, so put forward for inclusion
Orange = 70% or more of one group only rated outcome as 7–9, so put forward for discussion at consensus meeting.
Red= fewer than 70% of respondents in each group rated the outcome as 7–9, so outcome not put forward for inclusion.
CVI, cerebral visual impairment; 3D, 3 dimensional; TV, television.

on to discussing the rankings for all included outcomes and agreed a full list for the COS, after combining related outcomes to reduce overlap. Finally, attendees were asked to indicate their own personal 'top 10' outcomes for inclusion. A shorter COS list was agreed containing all outcomes that at least one person selected for their personal top 10. With permission, the consensus meeting discussions were recorded.

The Protocol for the development of this COS is included in online supplemental materials.

## RESULTS

We followed our protocol, but timing was later due to delays starting. For this reason we conducted 2 not 3 rounds of the Delphi survey.

### Stage 1: literature review

A total of 5155 titles and abstracts were extracted from four databases. These were searched for evaluative studies and 22 were selected for full text review. Of these, 13 papers (10 studies) met the inclusion criteria and data were extracted. Three papers[15–17] reported on the use of bifocal spectacles for children with reduced accommodation and used visual outcomes (near vision, accommodative amplitude, preferred working distance) and educational outcomes (reading tests). Three papers[18–20] reported outcomes after treatments for symptomatic convergence insufficiency and they used visual (nearpoint of convergence, ability to fuse images, symptom score), educational (reading tests) and behavioural outcomes (child behaviour checklists). Two papers investigated the effects of treatments for children with reduced visual attention in the context of a diagnosed attention deficit hyperactivity disorder: one used visual fields as the outcome[21] and one a score for visual attention.[22] Two papers investigated the effects of interventions to improve visuomotor skills: one involved prisms in children with hemispatial neglect and used visually guided pointing as the outcome[23] and one used eye movement training and used eye movement, arms movement and ball-catching accuracy as outcomes.[24] The remaining papers investigated the effects of intraventricular stem cell transplant on visual behaviour[25]; music on attention to task[26] and a parenting programme on parent behaviour,[27] all involving children with severe visual impairment. Thirty-seven individual outcomes were identified from the 13 papers in the review.

### Stage 2: qualitative interviews with families where a child has CVI

Families were from England (n=16), Scotland (n=1) and Northern Ireland (n=2) and one family lived in Eire. Twenty-four interviews were completed: 18 with parents/carers and 6 with children with a parent also present. Eight parent/carer interviews took place by telephone, 10 interviews were face to face.

Overall, CYP's ages ranged from 3 to 17 years. Ten children were under the age of 11, 6 were between 11 and 16 years of age and one was aged 17 years. The six children who took part in separate interviews were aged 6–17 years (two were in the younger group of 6–11 years and four were in the older group of 12–18 years). Of the six interviews conducted with CYP, one used video call, one was by telephone and four were face to face. Other diagnoses were recorded as described by parents and included premature birth, cerebral palsy; chromosomal abnormalities; epilepsy; global developmental delay and autism.

From the analysis, 22 outcomes were identified across four themes of: (1) assessment and understanding of implications of CVI for CYP and families, (2) education, (3) family life and (4) psychological well-being and quality of life.

All outcomes from the literature review and interviews were coded to the domains of the COMET taxonomy and these were used as section headings in the survey. We also included outcomes previously identified in a COS for children with neurodisability.[28] These were communication, emotional well-being, pain, sleep, mobility, self-care, independence, mental health, community and social life, behaviour, toileting and safety. Overlapping items were combined to reduce participant burden as much as possible. A final set of 23 discrete outcome domain items was included in the first round of the Delphi survey (table 1). After discussion to combine similar outcomes and remove duplicate items, the 23 outcomes included in the first round included 5 relating to visual functions that came from the proportionate review of the literature, 8 outcomes relating to general health and abilities from the neurodisability COS and 10 outcomes from the results of the qualitative interviews.

Table 3   Results from consensus meeting exercise: full list of items for inclusion in the core outcome set for childhood CVI and number of votes for participants' personal 'top 10', at consensus meeting

| No | Outcome | N who included outcome in their top ten (max 5) |
|----|---------|----|
| 1 | Community/social life and relationships* | 5 |
| 2 | Learning and accessing books or screens* | 5 |
| 3 | Child emotional well-being* | 4 |
| 4 | Tiredness/fatigue | 4 |
| 5 | Consistency of visual performance* | 4 |
| 6 | Managing in crowded environments* | 4 |
| 7 | Relevant adults aware of CVI* | 4 |
| 8 | Ability to deal with crowded or cluttered scene* | 4 |
| 9 | Mobility | 3 |
| 10 | Communication | 3 |
| 11 | Child's understanding of their own needs* | 3 |
| 12 | Safety | 2 |
| 13 | Environmental adaptations* | 2 |
| 14 | Ability to recognise visual targets | 1 |
| 15 | Family well-being | 1 |
| 16 | Ability to focus on near things | 0 |
| 17 | Ability to track moving objects | 0 |
| 18 | Child's Self-perception | 0 |
| 19 | Sleep | 0 |
| 20 | Self-care | 0 |
| 21 | Behaviour | 0 |
| 22 | Play | 0 |
| 23 | Functional 3D skills* | 0 |
| 24 | Sensitivity to light* | 0 |
| 25 | Ability to watch TV, films or run computer games* | 0 |
| 26 | Pain | 0 |
| 27 | Ability to cope with moving environment* | 0 |

*indicates outcomes suggested in full or in part by respondents during the Delphi survey or consensus meeting.
CVI, cerebral visual impairment; 3D, 3 dimensional.

## STAGE 3: DELPHI PROCESS

In total there were 126 participants in round 1 and 80 (64%) in round 2. The majority came from England (n=100), with the next largest group from Scotland (n=14) and small numbers from Wales (n=2), N Ireland (n=8) and Eire (n=2). They comprised QTVIs (29, then 21); consultant ophthalmologists (11 then 5); consultant paediatricians (4 then 2); orthoptists (8 then 5); habilitation officers (8 then 5) optometrists (3 then 2) and other professionals (17 then 12), plus the family members (46 then 28).

Twenty additional outcomes were suggested by participants in round 1 of the survey, some from CYP, some from adult family members and some from professionals. These included five vision-related outcomes including sensitivity to light, ability to see colour and to pick out features in crowded scenes; other sensory outcomes related to hearing; more outcomes relating to social functioning and family well-being and one outcome related to adults around the child understanding CVI.

A total of 43 outcomes were therefore ranked in round 2. Table 1 shows the initial list of 23 outcomes included for ranking by participants as well as the 20 additional outcomes suggested by participants, that were ranked in round 2.

Following the second round, the proportion of people ranking outcomes as of high importance with a ranking of 7–9 out of 9 was calculated. Of the total of 43 outcomes, 25 were 'in' initially as >70% of each stakeholder group had ranked them as 7–9. Table 2 summarises the ranking scores and shows outcomes in red where both groups ranked them lower than 70%, in green where both ranked higher than 70% and in orange where one group ranked higher and one lower.

### Stage 4: consensus workshop

Five people, two family representatives and three professionals, attended the consensus meeting in Bristol in March 2019. After a discussion, including of the nine outcomes where there was uncertainty, some similar items were combined and 27 outcomes were listed for inclusion in a final COS. Participants were then asked to vote on their ten most important outcomes. From this, a shorter COS list was produced, containing all the outcomes that at least one person put in their own top 10. The full list of all agreed outcomes is shown in table 3. The top 15 of these comprise a short COS, of outcomes at least one person put in their personal 'top 10'.

Figure 2 summarises the different stages of the process to identify the COS and the results obtained.

There were four vision-related outcomes in the shorter COS (consistency of visual performance, ability to pick out features in a cluttered scene, ability to cope in a crowded environment and ability to recognise visual targets) as well as the outcome that adults around the child understood about CVI. In the short COS, eight outcomes were suggested in part or completely, by the participants in round 1 (proposers included CYP, adult family members and professionals). Four were outcomes from the neurodisability COS. Two outcomes were voted as in their top 10 by all five attendees: 'community/interpersonal relations' and 'ability to learn including accessing books or screens'.

## DISCUSSION

This study brought together existing literature, families, professionals and us as researchers to reach a consensus on the most important outcomes to measure when conducting research into interventions to help children with CVI. There was good engagement by the participants at all stages and, although the consensus meeting numbers were small, a clear plan for the day and a committed group

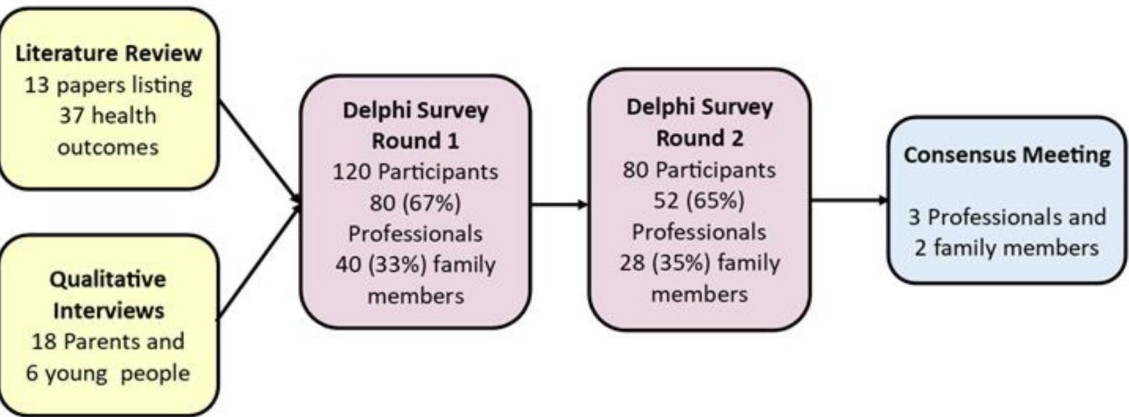

**Figure 2** Flow chart indicating the process involved in deriving the COS. COS, core outcome set.

of individuals were able to find common ground within their respective varied perspectives. Participants identified some outcomes with borderline results in round 2 as important to include and did not reject any outcomes that had been included on the basis of round 2 results. Although the participants were necessarily self-selecting and therefore not representative of the whole group or population with CVI, the consensus meeting validated, extended and ratified the Delphi survey results.

The COS of 27 items includes 11 vision-specific outcomes relating to a child's visual abilities, supporting the value of a dedicated COS for children with brain-related vision impairments, as opposed to using a more general COS for children with neurodisability. Outcomes included being able to see targets in cluttered scenes, or when moving and consistency of visual performance. These aspects of visual performance are asked about in several questionnaires relating to CVI[29 30] and the results of this exercise support the importance of asking about them when assessing a child with potential CVI. However, the majority of outcomes related to other domains in a child's life and their ability to engage with other people and their surroundings. Several of the outcomes in the full COS were originally identified in the COS for children with neurodisability, reflecting the range of comorbidities that are associated with CVI and were reported by our participants. The short COS featured 4 related to vision in real-world complex situations and the other 11 related to well-being, learning, interacting with the world and aspects of general health. This reflects the comments of the families in the interviews about the wide-ranging effects of CVI and indicates that professionals were also aware of the importance of non-visual outcomes.

This project was proportionate to the resources available and represents a first proposal of a COS for children with CVI, which is a relatively new field for clinical intervention studies. Our scope was the UK and Eire, and the children are likely have had a variety of visual problems as we did not have access to their medical records to check why or how they were diagnosed with CVI. However, presently, there is no agreed definition of CVI, beyond a consensus that the visual deficits involved

cannot be explained by ocular or optic nerve pathology. Future research may explore more targeted COS for children with CVI who display particular characteristics such reduced visual acuity or visuocognitive impairments or who fit defined CVI subtypes such as those recently described using a data-driven approach.[31]

Many of the outcomes are themselves broad, multi-faceted concepts such as 'Community, social life and relationships' and further work is needed to identify available validated measures for each outcome if available. However, the consistency with which these concepts were ranked as important indicates their importance to try and address, in some way. Increasing awareness of CVI was an important outcome for families and this may change as the condition becomes better understood and as agreed guidelines for diagnosis and management emerge.

Limitations of the study include that the literature review was targeted to specific study designs and some relevant evaluative studies may have been missed. We included papers reporting interventions targeting brain-related vision problems in children and may have included studies that some readers would not consider as relevant to CVI. We did not examine qualitative studies or health-related quality of life studies in children with CVI that may have elicited additional outcome domains. We did include an established COS for children with neurodisability, plus multiple items derived from the interviews, and we gathered additional suggestions from participants in the first round of survey. We are likely, therefore, to have included most or all outcomes of key importance and excluded those that were not felt to be important, in the views of our participants.

We might have identified more outcomes if we conducted more interviews, however, continuing to interview until no new themes emerge (data saturation) is not always appropriate[32] and we succeeded in our aim to elicit a rich set of outcomes with which to populate the first Delphi survey. There were few children who were interviewed and/or who took part in the Delphi survey and these were necessarily those who could self-report to a reasonable level. We may have under-represented the views of younger or less able children—however all had

 

been diagnosed with CVI and could speak to their experience about their lives. Only a small number of our participants were available to attend the consensus meeting which may have limited the range of opinions. Nevertheless all five participants were fully engaged and contributed to all aspects of the discussion and decision making. During the process, there were repeated exercises to reduce overlapping or similar types of outcome. This was a subjective process and may have resulted in loss of some more specific outcomes that in other settings would be rated important. There is still a degree of overlap between the outcomes and another group may have refined them differently. However, there was strong agreement that both visual function-related outcomes and child-centred outcomes relating to life chances and quality of life, were of crucial importance to include. We did not include users of COS such as researchers and journal editors, so did not capture their preferences in the consensus. We welcome discussion with colleagues to build on our proposed COS, which is offered as a useful starting point that may be refined in future, when national and/or international consensus is reached on the diagnosis of CVI in children.

A strength of this work is the high level of engagement of participants, the inclusion of children as well as parents and the broad levels of agreement between families and professionals. We used a variety of ways to identify relevant candidate outcomes and followed the COMET process that has been optimised iteratively over time. Several of the outcomes in the full COS and the short COS were suggested by the participants during the process. This indicates the value of inviting suggestions from participants to gain perspectives not suggested by the literature review or the interviews, perhaps because the context of rating outcomes stimulated different ideas from the more general discussion topics in the interviews.

## CONCLUSIONS

A first COS for intervention studies on childhood CVI has been proposed. More work is needed to identify which tools might best be used to measure these and some may have to be addressed qualitatively if no validated tools exist. Future COS research could also examine modifications of the full COS for more specified groups or interventions, that is, with a more restricted scope, or any changes that might result from including COS users, for example, journal editors or whether a having larger number of attendees at a consensus meeting leads to different selections of outcomes. Future research into interventions to help children with CVI should include at least the short COS items if possible, to increase relevance for the research for families and professionals and to aid cross-study comparisons.

**Acknowledgements** We are very grateful to the families and professionals who took part in this study, including the interviewees and participants in the Delphi survey and our consensus meeting: Kirsty Goodman, Janet Harwood, Kate Murton, Grainne Ni Dhuill and Penny Warnes. We are also grateful to the members of the PenCRU Family Faculty who advised.

**Contributors** The study was designed and led by CW, with help from all authors. CM advised on all aspects, CB did the literature search, TG led on the participant interviews and RW, TG and AP led on the Delphi study. AP drafted the first version of the manuscript and all authors contributed to revisions and approved the final manuscript.

**Funding** This research was funded by the National Institutes of Health Research (NIHR) as part of a Senior Research Fellowship for CW: SRF_2015_08_005.

**Disclaimer** The views expressed in this publication are those of the author(s) and not necessarily those of the NIHR, NHS or the UK Department of Health and Social Care.

**Competing interests** None declared.

**Patient consent for publication** Not applicable.

**Ethics approval** The study was approved by the University of Bristol Faculty of Health Sciences Research Ethics Committee (FREC ref: 58441).

**Provenance and peer review** Not commissioned; externally peer reviewed.

**Data availability statement** Data are available on reasonable request. Anonymised data will be available upon reasonable request.

**ORCID iDs**
Anna Pease http://orcid.org/0000-0002-3472-1047
Christopher Morris http://orcid.org/0000-0002-9916-507X
Cathy Williams http://orcid.org/0000-0002-9133-2021

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
