## [Reviewer comments · BMJ Open]

ARTICLE DETAILS

TITLE (PROVISIONAL)	Development of a core outcome set for evaluative research into Paediatric Cerebral Visual Impairment (CVI), in the UK and Eire
AUTHORS	Pease, Anna; Goodenough, Trudy; Borwick, Cath; Watanabe, Rose; Morris, Christopher; Williams, Cathy

VERSION 1 – REVIEW

REVIEWER	Chang, Melinda University of Southern California
REVIEW RETURNED	23-Apr-2021

GENERAL COMMENTS	This manuscript describes the development of a proposed core outcome set (COS) for evaluation of interventions for children with cerebral visual impairment (CVI). The authors are to be commended for attempting to address this difficult but important topic. Because CVI is a very heterogeneous condition, and there is no consensus as to how CVI should be defined, it is difficult to know what outcomes should be the focus of future trials. There are two major issues with the methodology of this study: 1) Due to absence of data substantiating the diagnosis of CVI in the patients whose families were invited to participate in the qualitative interviews and Delphi process, it is unclear how these results generalize to patients diagnosed with CVI by qualified practitioners; and 2) The literature review is, for the most part, not pertinent to the question being addressed. Issue 1: It is critical to know how patients were diagnosed with CVI and some basic characteristics including their visual acuity or other aspects of visual function and functional vision (e.g. CVI Range score). The authors state that they included "CVI as a broad concept, however diagnosed," but more details are necessary to demonstrate that children met diagnostic criteria for CVI accepted by at least some practitioners. Additionally, specifying the diagnostic criteria that were used, even if they were inconsistent, will help determine which patients this COS may be applicable to. The generalizability of the study is further compromised by the authors including only families of children older than 6 years and only 18 families in the qualitative interview process. Can the authors provide justification for these decisions? In the initial protocol, the plan was to interview 30 families or "continue until the point of diminishing returns." It seems difficult to believe that no new information was gathered after only 18 families were interviewed, given the heterogeneity of CVI. Is there prior literature to support interviewing what seems to be a rather small number of patients/families?
--

	Issue 2: Only one of the articles identified in the literature review is specific to CVI. There are two articles on bifocals for Down syndrome – some patients with Down syndrome do have CVI, but the accommodative insufficiency that may occur in these patients is not considered by most practitioners to be a symptom of CVI. There are several articles on convergence insufficiency (CI), a type of strabismus. Although the brainstem is believed to be involved in convergence, and some patients with CVI may have CI, CI is not generally considered a form of CVI. If we consider CI to be a manifestation of CVI, then any other type of strabismus would also meet criteria for CVI. Additional articles focus on attention deficit/hyperactivity disorder – although these patients have a neurodevelopmental disorder, deficits of visual attention in children with AD/HD are not typically considered a manifestation of CVI. If the authors wish to consider visual attention abnormalities in children with neurodevelopmental deficits as a type of CVI, then they would have to review the entire literature on children with autism who have deficits in social attention. The one article that actually focuses on children with CVI is the article on stem cell transplantation. In their literature search, the authors have neglected to include classic articles on the characteristics of CVI by Hoyt, Good, and colleagues, as well as more recent work by Gordon Dutton and Christine Roman-Lantzy. All of these studies describe important features of CVI that may be useful as part of a COS. Some of the features described in these studies (such as difficulty with visual crowding) were later identified by families during interviews and the Delphi process. If the authors are able to address these two major issues, this study could be an important contribution to the literature.
--	---

REVIEWER	Dutton, Gordon Glasgow Caledonian University, Vision Sciences
REVIEW RETURNED	29-Apr-2021

GENERAL COMMENTS	This paper describes a well-conceived and presented project. The authors have sought out and identified a preliminary key set of core adverse outcomes in children due to the broad spectrum of causative cerebral visual impairments and their associated conditions. From this referee's experience and perspective, each one of the elements of the core outcome sets identified by the authors, can potentially be traced back to each individual child's profile of specific cerebral visual impairments. There is good empirical evidence that matched targeted sets of specific interventions can prove effective in prevention and / or amelioration of these adverse outcomes. It can thus be cogently argued that the primary purpose of the project described was to identify a core outcome set, with the specific purpose of future development of evidence based strategies founded upon optimal choice of the currently employed empirical approaches. Although the final paragraph of the conclusion of this paper states that 'Future research into intervention to help children with CVI should include if possible, at least the short COS items, to increase relevance for the research for families and professionals
--

	and to aid cross-study comparisons', this concept framework is not mentioned elsewhere in the paper. It is apparent that the principal purpose of the project described, was to identify the core (adverse) outcomes, in order to work towards their optimal prevention and / or amelioration. Yet this is not highlighted as an aim. It can thus be argued that the final paragraph of the introduction warrants extension along these lines. Minor P33 Line 11: consistence of visual performance - should probably read – consistency of visual performance P13 Line 54: The word professional – should read - professionals P11 Line 1: Managing crowded environments - should arguably read - Managing in crowded environments', or 'Coping with crowded environments'.
--	--

VERSION 1 – AUTHOR RESPONSE

Reviewer: 1

Dr. Melinda Chang, University of Southern California

Comments to the Author:

This manuscript describes the development of a proposed core outcome set (COS) for evaluation of interventions for children with cerebral visual impairment (CVI). The authors are to be commended for attempting to address this difficult but important topic. Because CVI is a very heterogeneous condition, and there is no consensus as to how CVI should be defined, it is difficult to know what outcomes should be the focus of future trials.

There are two major issues with the methodology of this study: 1) Due to absence of data substantiating the diagnosis of CVI in the patients whose families were invited to participate in the qualitative interviews and Delphi process, it is unclear how these results generalize to patients diagnosed with CVI by qualified practitioners; and 2) The literature review is, for the most part, not pertinent to the question being addressed.

We thank the reviewer for raising these issues and we discuss them below.

Issue 1: It is critical to know how patients were diagnosed with CVI and some basic characteristics including their visual acuity or other aspects of visual function and functional vision (e.g. CVI Range score). The authors state that they included “CVI as a broad concept, however diagnosed,” but more details are necessary to demonstrate that children met diagnostic criteria for CVI accepted by at least some practitioners. Additionally, specifying the diagnostic criteria that were used, even if they were inconsistent, will help determine which patients this COS may be applicable to.

We agree with the reviewer that this is an important topic. We have now emphasised that the criteria used to diagnose CVI vary within and between countries. As this is a first COS for this patient group, and in the absence of agreed diagnostic criteria, we present our findings as a starting point for clinicians and researchers, based on the reports of a group of children and families with a likely wide range of visual difficulties. All however were diagnosed with “CVI” by a professional and we suggest that until we have agreed criteria, there will inevitably be variations in the clinical profiles of children with this diagnostic label.

The generalizability of the study is further compromised by the authors including only families of children older than 6 years and only 18 families in the qualitative interview process. Can the authors provide justification for these decisions? In the initial protocol, the plan was to interview 30 families or “continue until the point of diminishing returns.” It seems difficult to believe that no new information

was gathered after only 18 families were interviewed, given the heterogeneity of CVI. Is there prior literature to support interviewing what seems to be a rather small number of patients/families? We have included a comment that with more interviews we might have generated more outcomes with which to fill the first Delphi, but we did elicit a rich source of outcomes important to families and further outcomes were suggested from the participants. We have quoted a review arguing that data saturation may not be an important concept for all forms of thematic analysis. We have also removed the line in the methods saying that we continued until the point of diminishing returns.

Issue 2: Only one of the articles identified in the literature review is specific to CVI. There are two articles on bifocals for Down syndrome – some patients with Down syndrome do have CVI, but the accommodative insufficiency that may occur in these patients is not considered by most practitioners to be a symptom of CVI. There are several articles on convergence insufficiency (CI), a type of strabismus. Although the brainstem is believed to be involved in convergence, and some patients with CVI may have CI, CI is not generally considered a form of CVI. If we consider CI to be a manifestation of CVI, then any other type of strabismus would also meet criteria for CVI. Additional articles focus on attention deficit/hyperactivity disorder – although these patients have a neurodevelopmental disorder, deficits of visual attention in children with AD/HD are not typically considered a manifestation of CVI. If the authors wish to consider visual attention abnormalities in children with neurodevelopmental deficits as a type of CVI, then they would have to review the entire literature on children with autism who have deficits in social attention. The one article that actually focuses on children with CVI is the article on stem cell transplantation.

In their literature search, the authors have neglected to include classic articles on the characteristics of CVI by Hoyt, Good, and colleagues, as well as more recent work by Gordon Dutton and Christine Roman-Lantzy. All of these studies describe important features of CVI that may be useful as part of a COS. Some of the features described in these studies (such as difficulty with visual crowding) were later identified by families during interviews and the Delphi process.

If the authors are able to address these two major issues, this study could be an important contribution to the literature.

Again we thank the reviewer for raising these important issues. We have included in the introduction more of the seminal works describing CVI, and we agree these are helpful. The proportionate literature review was focussed on studies evaluating interventions, not descriptive studies, reflecting the scope of our COS. We have made this clearer in the Title, Methods and Discussion, as this lack of clarity was also raised by Reviewer 2. Our inclusion criteria specified controlled or before-and-after evaluative studies and again we aimed to be inclusive of brain-related vision problems, reflecting the lack of agreement about the diagnosis of CVI. We have included a comment, agreeing with the reviewer, that some clinicians would not regard some of the topics of included papers as being part of CVI. However, as with the interviews, the aim was to populate the first Delphi survey with sample outcomes for the participants to rate and it was notable that several outcomes missing from the first survey, were added in by participants, reflecting their importance to the stakeholders.

Reviewer: 2

Dr. Gordon Dutton, Glasgow Caledonian University

Comments to the Author:

This paper describes a well-conceived and presented project.

The authors have sought out and identified a preliminary key set of core adverse outcomes in children due to the broad spectrum of causative cerebral visual impairments and their associated conditions. From this referee's experience and perspective, each one of the elements of the core outcome sets identified by the authors, can potentially be traced back to each individual child's profile of specific cerebral visual impairments. There is good empirical evidence that matched targeted sets of specific

interventions can prove effective in prevention and / or amelioration of these adverse outcomes. It can thus be cogently argued that the primary purpose of the project described was to identify a core outcome set, with the specific purpose of future development of evidence based strategies founded upon optimal choice of the currently employed empirical approaches.

Although the final paragraph of the conclusion of this paper states that 'Future research into intervention to help children with CVI should include if possible, at least the short COS items, to increase relevance for the research for families and professionals and to aid cross-study comparisons', this concept framework is not mentioned elsewhere in the paper.

It is apparent that the principal purpose of the project described, was to identify the core (adverse) outcomes, in order to work towards their optimal prevention and / or amelioration. Yet this is not highlighted as an aim. It can thus be argued that the final paragraph of the introduction warrants extension along these lines.

We thank the Reviewer for this important point and have emphasised in the Introduction and the Discussion, that as the Reviewer says, the whole aim of this COS is ultimately to lead to improved outcomes for children with CVI, by eliciting the range of issues in their lives that can be adversely affected by the condition, so that future evaluative work can investigate the effectiveness of strategies to improve those issues.

Minor

P33 Line 11: consistence of visual performance - should probably read – consistency of visual performance

Thank you we have corrected this

P13 Line 54: The word professional – should read – professionals

Thank you we have corrected this

P11 Line 1: Managing crowded environments - should arguably read - Managing in crowded environments', or 'Coping with crowded environments'.

Thank you we have corrected this

Reviewer: 1

Competing interests of Reviewer: none

Reviewer: 2

Competing interests of Reviewer: None

VERSION 2 – REVIEW

REVIEWER	Chang, Melinda University of Southern California
REVIEW RETURNED	02-Sep-2021

GENERAL COMMENTS	The two issues identified in the prior version of the manuscript remain: the lack of clarity on the clinical characteristics of the children with CVI whose families were included in this study, and the literature review primarily including studies of children who had disorders other than CVI. However, the authors have pointed out these limitations in the revised manuscript.
--